# Molecular Mechanism Contributing to Malnutrition and Sarcopenia in Patients with Liver Cirrhosis

**DOI:** 10.3390/ijms21155357

**Published:** 2020-07-28

**Authors:** Fatuma Meyer, Karen Bannert, Mats Wiese, Susanne Esau, Lea F. Sautter, Luise Ehlers, Ali A. Aghdassi, Cornelia C. Metges, Leif-A. Garbe, Robert Jaster, Markus M. Lerch, Georg Lamprecht, Luzia Valentini

**Affiliations:** 1Department of Agriculture and Food Sciences, Neubrandenburg Institute for Evidence-Based Dietetics (NIED), University of Applied Sciences Neubrandenburg, 17033 Neubrandenburg, Germany; fmeyer@hs-nb.de (F.M.); esau@hs-nb.de (S.E.); leafranziska.sautter@med.uni-rostock.de (L.F.S.); 2Division of Gastroenterology and Endocrinology, Department of Internal Medicine II, University Medicine Rostock, 18057 Rostock, Germany; karen.bannert@med.uni-rostock.de (K.B.); luise.ehlers@med.uni-rostock.de (L.E.); robert.jaster@med.uni-rostock.de (R.J.); georg.lamprecht@med.uni-rostock.de (G.L.); 3Division of Gastroenterology, Endocrinology and Nutritional Medicine, Department of Internal Medicine A, University Medicine Greifswald, 17475 Greifswald, Germany; mats.wiese@med.uni-greifswald.de (M.W.); ali.aghdassi@med.uni-greifswald.de (A.A.A.); lerch@uni-greifswald.de (M.M.L.); 4Institute of Nutritional Physiology ‘Oskar Kellner’, Leibniz Institute for Farm Animal Biology (FBN), 18196 Dummerstorf, Germany; metges@fbn-dummerstorf.de; 5Department of Agriculture and Food Sciences, University of Applied Sciences Neubrandenburg, 17033 Neubrandenburg, Germany; garbe@hs-nb.de

**Keywords:** cirrhosis, malnutrition, sarcopenia, protein turnover, hypermetabolism, hyperammonemia, myostatin, growth hormone

## Abstract

Liver cirrhosis is frequently accompanied by disease-related malnutrition (DRM) and sarcopenia, defined as loss of skeletal muscle mass and function. DRM and sarcopenia often coexist in cirrhotic patients and are associated with increased morbidity and mortality. The clinical manifestation of both comorbidities are triggered by multifactorial mechanisms including reduced nutrient and energy intake caused by dietary restrictions, anorexia, neuroendocrine deregulation, olfactory and gustatory deficits. Maldigestion and malabsorption due to small intestinal bacterial overgrowth, pancreatic insufficiency or cholestasis may also contribute to DRM and sarcopenia. Decreased protein synthesis and increased protein degradation is the cornerstone mechanism to muscle loss, among others mediated by disease- and inflammation-mediated metabolic changes, hyperammonemia, increased myostatin and reduced human growth hormone. The concise pathophysiological mechanisms and interactions of DRM and sarcopenia in liver cirrhosis are not completely understood. Furthermore, most knowledge in this field are based on experimental models, but only few data in humans exist. This review summarizes known and proposed molecular mechanisms contributing to malnutrition and sarcopenia in liver cirrhosis and highlights remaining knowledge gaps. Since, in the prevention and treatment of DRM and sarcopenia in cirrhotic patients, more research is needed to identify potential biomarkers for diagnosis and development of targeted therapeutic strategies.

## 1. Introduction

Liver diseases account for approximately 2 million deaths worldwide per year with about half the numbers due to complications related to liver cirrhosis [1,2]. Liver cirrhosis can lead to many complications ending up in high mortality [3] and according to the World Health Organisation, in 2015, most viral hepatitis deaths were due to chronic liver disease with 720,000 deaths due to liver cirrhosis [4]. In Germany, more than 62,000 patients with chronic liver diseases were admitted to hospital in 2016 [5] showing an increasing trend and imposing a high medical and economic burden. Liver cirrhosis is the late and irreversible stage of hepatic fibrosis [6], pathologically representing a hepatic tissue remodeling with the formation of fibrotic interconnected septa that distort the liver tissue, divide the parenchyma in nodules and alter vascular architecture and resistance resulting in altered blood flow and portal hypertension [7]. The most common underlying etiologies of liver cirrhosis are alcohol abuse, chronic hepatitis B virus (HBV) and hepatitis C virus (HCV) infections, as well as non-alcoholic steatohepatitis (NASH) [8]. Based on geographical location, the etiology of liver cirrhosis varies with HBV being the primary cause in the Asia-Pacific region while in Europe, increase in liver cirrhosis may be linked to alcohol abuse in Northern Europe and viral hepatitis epidemics in Eastern and Southern European countries [9].

The liver is the second largest organ in the human body [10] and primary organ for metabolism [11]. Its functional integrity is essential for the supply and inter-organ trafficking of macronutrients (proteins, fat and carbohydrates) and their metabolism [12,13]. Nutrition is important in the management of liver cirrhosis and its complications [14]. Given the closely intertwined relation between the liver and nutrition, malnutrition is a very frequent feature in patients with liver cirrhosis [13,15,16,17,18].

### Disease-Related Malnutrition and Sarcopenia

Disease-related malnutrition (DRM) describes a nutrition- and inflammation-related disorder that results from prolonged acute or chronic disease and lack of nutrient intake or absorption leading to compromised body composition and function. The hallmarks of malnutrition in liver cirrhosis are loss of muscle mass and -function with or without loss of subcutaneous adipose tissue [19,20,21]. According to the Global Leadership Initiative on Malnutrition (GLIM), the diagnosis of DRM requires a combination of phenotypical (unintentional loss of body weight, low body-mass index (BMI) or reduced muscle mass) and etiological criteria (reduced food intake/assimilation, inflammation or disease burden) [22]. DRM is very common among patients with liver cirrhosis occurring in at least 50% and up to 90% of patients and specifically high among patients with decompensated liver cirrhosis, while exceeding one-fifth of those with compensated liver cirrhosis [14,21,23,24]. This wide range is accounted for by the variation in methods used for the assessment of malnutrition [20]. In a recent study with 170 cirrhotic patients, the prevalence of malnutrition risk varied according to the screening tools used ranging from 13.5 to 54.1% [25].

DRM is a robust and independent predictor of adverse clinical outcomes including infections, hepatic encephalopathy, hospitalization and mortality [21,26]. In many patient populations, DRM and sarcopenia are present in parallel, and often manifest clinically in a malnutrition-sarcopenia syndrome (MSS) [27]. In a study by Kalafateli and colleagues [28], predicting post liver transplantation outcomes, showed that DRM and sarcopenia are independent predictors of longer mechanical ventilation, longer intensive care unit and hospital stay, increased incidence of infections, and 12-month mortality post liver transplantation.

Skeletal muscle loss is a principal component of malnutrition in liver disease [29,30]. Loss of lean body mass negatively affects the clinical course of acute cirrhotic complications with resultant poor prognosis [31]. In liver cirrhosis, skeletal muscle abnormalities are a significant and a major factor of DRM affecting 30%–70% of patients [15,32,33,34]. The progressive and diffuse loss of skeletal muscle strength, mass as well as decreased functional capacity is more defined and diagnosed in detail by sarcopenia [15,32,35]. According to the European Working Group on Sarcopenia in Older People (EWGSOP2), sarcopenia is diagnosed by the presence of both low muscle strength and low muscle quantity or quality with poor physical performance as indicative of severe sarcopenia [36,37]. Skeletal muscle volume is reported to have a strong correlation with physical performance, such as gait speed, as well as with muscle strength, such as grip strength [38]. Severe sarcopenia in patients with liver cirrhosis is related to increased complications both pre and post liver transplantation [28,39].

Despite the prognostic significance of DRM and sarcopenia in chronic liver disease, diagnosis is difficult and the concise pathophysiological mechanisms are not completely understood [39,40]. Presently, there are no validated objective biomarkers for identification and monitoring of DRM available [40]. Furthermore, sarcopenia in liver cirrhosis cannot completely be explained by simple DRM, it is difficult to treat, and there are currently no proven effective therapies to prevent or reverse sarcopenia [32]. However, the knowledge concerning the molecular mechanisms involved in muscle wasting has increased, mainly following studies on experimental models with few data concerning cirrhotic patients [41]. With most therapies to date having focused on the principle of deficiency replacement, rather than targeted treatments, and since sarcopenia and DRM are major contributors of clinical outcomes, understanding the biochemical and cellular mechanisms that result in loss of muscle mass is critical in the identification of appropriate therapeutic targets [42].

This article provides an overview of the possible molecular mechanisms contributing to DRM and sarcopenia in patients with liver cirrhosis. The main objective is to summarize the current understanding of the molecular mechanisms in relation to the clinical problems, to highlight knowledge gaps and strategies to identify molecular targets for future therapeutic interventions.

## 2. Molecular Mechanisms Contributing to Disease-Related Malnutrition (DRM) and Sarcopenia in Liver Cirrhosis

The mechanisms of DRM and sarcopenia in liver cirrhosis are multifactorial, complex and not completely understood [2,43]. These include decreased nutrient intake (both energy and protein), intestinal maldigestion and/or malabsorption and hypercatabolism among others have been identified [19,21,44]. Frequent fasting states and external factors, such as alcohol, infections and medical drugs further contribute to DRM in patients with liver cirrhosis [19]. Leading to muscle depletion, sarcopenia in liver cirrhosis is affected by DRM, alterations in protein turnover, energy disposal, hormonal and metabolic changes, increased myostatin expression, reactive oxygen species and inflammatory cytokines [11,32,45]. Although the pathogenesis of sarcopenia in liver cirrhosis is poorly understood, a number of proposed mechanisms have been previously described representing an imbalance between muscle breakdown and formation [32]. Muscle mass is preserved by an equilibrium between protein synthesis, protein breakdown and regenerative capability. In liver cirrhosis, decreased energy intake further aggravates impaired muscle biosynthesis and increased muscle proteolysis caused by low glycogen stores resulting in an increased need for gluconeogenesis [20,42,46]. There are several factors associated with sarcopenia in liver cirrhosis including hyperammonemia, low testosterone levels, decreased human growth hormone (GH) and high endotoxin levels among others [20]. The mechanisms contributing to malnutrition and sarcopenia in liver cirrhosis are summarized in Figure 1 and detailed in Figure 2 as discussed in the following sections.

### 2.1. Energy-Associated Mechanisms

#### 2.1.1. Reduced Nutrient and Energy Intake

Impaired dietary intake is a principal cause of DRM and sarcopenia [44]. In liver cirrhosis, several factors contribute to reduced oral intake including anorexia, decreased sense of smell and dysgeusia, nausea, abdominal pain and bloating. These may be worsened or even caused by micronutrient deficiencies such as zinc and magnesium [2,15,47,48]. The majority of cirrhotic patients unintentionally follow a low energy diet [49]. Reduced energy intake has been linked to a higher prevalence of sarcopenia, limited regeneration capacity and prolonged functional recovery of the liver [47,50,51]. Frequent *nil per os* prescription in inpatients required for diagnostic procedures can contribute to DRM and sarcopenia [2,19].

Anorexia is grossly defined as the loss of the desire to eat and is almost invariably associated with reduced food intake with severe deterioration of patient’s nutritional status [52,53]. Contributing to DRM, anorexia is a major symptom associated with liver cirrhosis and of paramount importance [29,54] as it manifests with severe malnutrition and organ failure [55]. Similar to DRM, in sarcopenia, barriers to adequate calorie and protein intake such as anorexia and nausea are also existent [32]. In patients with decompensated liver cirrhosis, anorexia contributes up to an 80% prevalence of DRM [56] while 90% of patients with advanced alcoholic liver disease experience anorexia [10]. Anorexia may be triggered by an imbalance between orexigenic and anorexigenic hormones, by the chronic increase in circulating proinflammatory cytokines, zinc and vitamin A deficiency, delayed gastric emptying and portal hypertension resulting in gastric and intestinal tissue congestion as well extrinsic compression due to ascites [15,32,44,49,57]. Anorectic patients present with early satiety, smell and taste disorders and meat aversion as well as nausea/vomiting [53].

##### Regulators of Appetite and Satiety

Neuroendocrine regulation of satiety and hunger is complex and involves one major orexigenic (appetite) hormone, ghrelin, and a number of anorexigenic (satiety) hormones including leptin, cholecystokinin (CCK), glucagon like peptide-1 (GLP-1), peptide YY (PYY), oxyntomodulin and pancreatic polypeptide (PP) [56,58,59].

##### Ghrelin

Ghrelin, which is produced mainly by enteroendocrine cells in the stomach, is the only circulating gastrointestinal hormone that has orexigenic effects [60]. Ghrelin levels increase abruptly before the onset of a meal and decrease rapidly after eating suggesting it signals meal initiation [43,60] and stimulates food intake [61]. Ghrelin controls energy balance, enhances fat mass deposition and food intake through the activation of the hypothalamic nuclei as well as the promotion of neuropeptide Y (NPY) and agouti-related protein (AGRP) expression, the latter being one of the most potent and long-lasting appetite stimulators [50]. Recent data suggest direct effects of ghrelin on hepatic stellate cells, resulting in diminished liver fibrosis [62]. In a study investigating the relation of basal and postprandial concentrations of ghrelin, leptin, plasma glucose and insulin to energy intake and resting energy expenditure, patients with liver cirrhosis showed lower ghrelin concentrations, higher postprandial glucose and elevated baseline leptin levels [43].

##### Leptin

Leptin is an anorexigenic hormone that exerts part of its effects by inhibiting orexigenic neurons and activating anorexigenic neurons in the hypothalamus [63]. Formally, it belongs to the adipokines, a group of endocrine peptide hormones secreted from the adipose tissue [50,64]. Leptin is a 16 kDa cytokine-like peptide hormone with a α-helix tertiary structure similar to the cytokines interleukin-6 (IL-6) and IL-12 [65]. It is encoded by the *ob* gene and has a well-established key function in maintaining body weight [65,66,67]. Leptin plays a fundamental role in understanding the control of energy balance [53]. It not only suppresses appetite but also increases energy expenditure, whereby the mechanism of action in the hypothalamic nuclei is antagonistic to ghrelin; in that leptin inhibits NPY and AGRP expression [50]. Serum concentrations of leptin correlate with body fat mass and are quickly reduced by fasting [67]. Patients with liver cirrhosis show increased serum leptin levels, particularly when expressed in unit fat mass [18,43,65], with a 2-fold increase in fasting levels compared with healthy individuals [68]. Increase in serum leptin concentration has also been linked to early satiety suggesting its pathophysiological role in the development of anorexia in this patient population [69].

##### Cholecystokinin (CCK), Glucagon Like peptide-1 (GLP-1), Peptide YY (PYY)

CCK is expressed by the enteroendocrine I cells in the duodenum und jejunum [70]. It reduces food intake in animals and humans and delays gastric emptying. Similarly, GLP-1 also delays gastric emptying; however, the primary action is to stimulate insulin secretion [60]. GLP-1 is produced by the L-cells, a group of enteroendocrine cells in the ileum and colon. PYY is a 36–amino acid hormone related to neuropeptide Y (NPY) [56] and is also secreted by L-cells [60]. Similar to CCK and GLP-1, PYY is secreted into the circulation in proportion to meal size [56]. PYY inhibits gastrointestinal motility, suppresses pancreatic secretion and also delays gastric emptying [71]. It also binds to all subtypes of the NPY family receptors. PYY_3-36_ is the 34-amino acid enzymatic breakdown product of PYY and major form in human serum. PYY_3-36_ has a strong affinity to the Y2 receptors and in humans decreases appetite as well as energy intake [60]. Increased fasting serum levels of PYY_3-36_ were found in patients with decompensated liver cirrhosis, while baseline PYY_3-36_ was normal in patients with compensated liver cirrhosis and controls [56]. After oral nutritional stimulus, the expected increase in PYY_3-36_ was only observed in controls and in patients with compensated liver cirrhosis but not in decompensated liver cirrhosis. This points towards an enteroendocrine dysregulation in decompensated liver cirrhosis [43].

##### Proinflammatory Cytokines

As previously mentioned, anorexia also results from proinflammatory activity and has both central and peripheral elements with loss of appetite being attributable to the presence of cytokines [2,15,32,47,49]. Liver cirrhosis is a known proinflammatory condition as represented by elevated levels of tumor necrosis factor-α (TNF-α), interleukin-1β (IL-1β), and IL-6 [32,72]. These and other proinflammatory cytokines are also notorious anorexic mediators [73]. In liver cirrhosis, the activation of the pro-inflammatory state is probably caused by the translocation of bacterial products, which are recognized as pathogen associated molecular patterns (PAMPs) by immune cells as a result of portal hypertension and dysbiosis that compromise the gut barrier function [15]. Proinflammatory cytokines trigger DRM in several ways; they are inversely related to nutrient intake, potentially decrease appetite and can contribute to hypermetabolism [2]. In liver cirrhosis circulating proinflammatory cytokines are associated with alterations in gut microbiome and compromised epithelial gut barrier function [74], as evidenced by endotoxemia, i.e., increased bacterial lipopolysaccharides (LPS) concentrations, in the portal and/or systemic blood [75]. Hepatocellular and immune dysfunction as well as portosystemic shunting worsen the endotoxemia which via TNF-α-dependent and independent pathways may also lead to increased protein breakdown via autophagy and reduced protein synthesis [15]. Endotoxemia in small intestinal bacterial overgrowth (SIBO) patients likely activate the toll-like receptor 4 (TLR-4) and cluster of differentiation 14 (CD14) receptor by stimulating the expression of nuclear factor-κB (NF-κB) that mediates the production of proinflammatory cytokines [76].

##### Regulators of Taste and Smell

Olfactory deficits and dysgeusia are frequently described in chronic liver diseases and might contribute to decreased macro- and micronutrient intake leading to nutrient deficits [77]. For example, vitamin A and zinc deficiency may be the cause of an altered sense of taste observed in a proportion of patients with advanced liver disease, also partly due to neurotoxins or an abnormal excretion of sulfur metabolites [78]. Hypozincemia, or zinc deficiency in liver disease, is associated with the development of alterations in taste and smell as well as anorexia, which can further contribute to a decreased food intake as well as increased gastrointestinal and urinary losses [23]. In patients with liver cirrhosis, the two most common symptoms related to taste are sweet taste aversion and salty taste intolerance, but also a metallic taste is frequently observed [79], whereby patients with hypozincemia report having either a dry mouth or a metallic taste [23]. The urinary loss of zinc is aggravated by the prevalent use of diuretics used to treat edema and ascites [80]. Meanwhile, medications used for oral treatment can also lead to taste alterations or even associated side effects, such as nausea and flatulence, consequently leading to appetite loss [52].

##### Dietary Restrictions

Further contributing to less palatable diets in patients with liver cirrhosis are frequent dietary restrictions, appropriate in the case of sodium limitations but often inappropriate in the case of protein restriction [2,47]. Suggested in earlier dietary recommendations for cirrhotic patients is a restricted protein diet. Although this is currently no longer recommended, it is unfortunately widely practiced and consequently leads to protein deficiency [23]. A major complication of decompensated liver cirrhosis is hepatic encephalopathy with an incidence of 30%–50% [81]. Protein restriction during acute exacerbations of hepatic encephalopathy is no longer recommended, as it increases protein catabolism and does not show any clinical advantage [82,83]. Hepatic encephalopathy on its own can contribute to anorexia, difficulty in swallowing and/or chewing, limited access to food and poor appetite [2,57]. In terms of appropriate restrictions, cirrhotic patients are advised to follow a low sodium diet if they have decompensated disease with ascites [84]. However, adherence to a low sodium diet is often difficult because of the bland taste, that subsequently leads to low energy intake. As demonstrated by Morando et al. [85], cirrhotic patients on a low sodium diet had a 20% reduced mean daily energy intake compared with patients not following the diet.

#### 2.1.2. Maldigestion and Malabsorption

Malabsorption is another important mechanism leading to malnutrition and sarcopenia [78]. There are multiple factors that result in nutrient malabsorption in patients with liver cirrhosis [44]. These include pancreatic insufficiency due to chronic alcoholic pancreatitis, cholestasis and drug-related diarrhea (lactulose, antibiotics, diuretics, cholestyramine). Chronic alcohol abuse may result in concomitant pancreatic insufficiency which can also impair nutrient absorption [47,86]. It is also possible that alcohol, rather than the underlying liver pathology, could be an essential variable in DRM in liver disease [29].

##### Cholestasis

Cholestasis can also result in malabsorption of nutrients [87]. Cholestasis is caused by a disruption of bile flow which results in a lack of bile in the intestine, accumulation of toxic bile acids and other metabolites in the liver, and increased bile acids in the systemic circulation [88]. Cholestatic liver diseases, such as primary biliary cholangitis (PBC) and primary sclerosing cholangitis (PSC) cause a dysfunction in bile acid metabolism and secretion which can result in malabsorption of fat and fat-soluble vitamins [29]. Especially in malnourished liver cirrhosis patients, fat malabsorption has been frequently reported [89]. Disturbances in bile acid metabolism affect the formation of micelles [2] and the absorption of long-chain fatty acids through the lymphatic route [78] which are necessary for fat digestion and absorption of fat-soluble vitamin [2]. Lack of fat-soluble vitamins is a common manifestation in cirrhotic patients due to both poor oral intake and malabsorption [13,87].

##### Small Intestinal Bacterial Overgrowth and Gut Microbiome

In patients with advanced liver disease an additional mechanism associated with malabsorption and DRM is small intestinal bacterial overgrowth (SIBO). SIBO is a condition in which colonic bacteria colonize the small bowel and impair microvilli function, digestive enzyme production and intestinal barrier dysfunction causing disturbed absorption and metabolism of nutrients and affecting intestinal motility [47,90,91,92]. The most common symptoms include abdominal pain, diarrhea, flatulence and abdominal overflow [90]. SIBO was found in 61% of cirrhotic patients based on small intestinal cultures [2,93] and it may be involved in bacterial translocation and infectious complications, such as spontaneous bacterial peritonitis [89].

Moreover, an impaired exocrine pancreatic function in alcoholic liver cirrhosis is associated with alterations of the gut microbiome that have a stronger effect than alcohol consumption and other dietary factors [94]. Changes in the gut microbiome are more evident in patients with decompensated liver cirrhosis [95].

#### 2.1.3. Altered Macronutrient Metabolism

Altered macronutrient metabolism is a cornerstone mechanism contributing to DRM in liver cirrhosis [44]. Glucose metabolism has been well studied in liver disease with cirrhotic patients having increased levels of gluconeogenesis, protein catabolism and decreased levels of glycogenolysis compared with healthy individuals leading to significant depletion of protein and fat reserves, reported in almost 50% of patients with liver cirrhosis [68,78]. Several factors contribute to increased rates of gluconeogenesis. First of all, as a result of liver cirrhosis, the ability of hepatocytes to store, synthesize and break down glycogen is reduced. These defects promote gluconeogenesis from protein as alternative fuel source [68]. Due to a decrease in glycogen stores and increased gluconeogenesis, energy metabolism shifts from carbohydrate to fat oxidation while insulin resistance may also develop [12,19,57]. Tissues that are glucose dependent will need gluconeogenesis from amino acids since fatty acids cannot be used for this process. This in turn leads to break down of glycogenic amino acids from the skeletal muscles so that the adequate amount of glucose can be produced. In cirrhotic patients frequent and repeated fasting results in recurrent proteolysis triggered by the amino acid needs for gluconeogenesis and resulting in muscle loss consequently sarcopenia [15,57]. Following a short overnight fast, the rate of fat and protein catabolism in patients with cirrhosis is similar to that of healthy subjects who underwent 2 to 3 days of starvation [68]. These results are confirmed by indirect calorimetry showing decreased carbohydrate oxidation and increased fat oxidation in the early morning hours in cirrhotic patients [96,97].

##### Glucose Metabolism

Patients with liver cirrhosis often display insulin resistance with 60%–80% having impaired glucose tolerance (IGT) while 10%–15% developing overt diabetes [72]. In patients with decompensated liver cirrhosis, abnormal glucose tolerance is an independent predictor of 30-day mortality [98]. Hepatogenous insulin resistance is characterized mainly by peripheral insulin resistance in the skeletal muscle and fat tissue, while uptake of glucose in the liver is normal or even enhanced [98]. The mechanisms accounting for insulin resistance in liver cirrhosis are still largely unknown [99]. Contributing factors may include altered glycolytic enzyme activity, changes in specific glucose transporter expression or impairment of membrane receptors for insulin [100] as well as decreased first past uptake of insulin in the liver due to liver damage and shunting of blood into the systemic circulation. Insulin resistance decreases peripheral glucose utilization and contributes to decreased hepatic glucose production and hepatic glycogen reserves, although, increased serum levels of glucagon, which result from impaired degradation by the liver, increase the rate of gluconeogenesis [68].

##### Alterations in Protein Turnover

As mentioned previously, the progression of chronic liver disease results in gluconeogenesis that requires a higher protein intake than usual [101,102]. Metabolic changes and alterations in protein turnover are major factors in muscle depletion in sarcopenia of chronic disease [32]. In decompensated liver cirrhosis, the catabolic state is characterized by an imbalance of two groups of plasma amino acids, decreased branched-chain amino acids (BCAAs: valine, leucine, isoleucine) and increased aromatic amino acids (AAAs: tyrosine, phenylalanine, tryptophane) [103]. The primary source of amino acids for gluconeogenesis is proteolysis in the skeletal muscle that generates both BCAA and AAA. However, only BCAA are catabolized in the skeletal muscle due to the localization of the branched chain ketodehydrogenase [42]. In liver cirrhosis progression, the depletion of BCAAs inhibit protein synthesis and protein turnover [101,102]. This ultimately leads to the aforementioned catabolism and increased muscular atrophy [104]. Infection can increase rates of protein catabolism, whereby, the production of cytokines and other infection mediators activate proteolysis and increase oxidation of BCAAs [68].

##### Hypermetabolism

DRM might also contribute to increased resting energy expenditure (REE) observed in 15%–34% of patients with liver cirrhosis [2,57,78]. In order to maintain nutritional equilibrium, energy supply must balance total energy expenditure (TEE) [45]. REE is the amount of energy an individual uses to perform vital organ functions, free of activity and digestion [68]. In cirrhotic patients, REE varies depending on medical condition, presence of inflammation, and degree of malnutrition [15,19,45]. This hyperdynamic circulation leads to systemic vasodilation and an expanded intravascular blood volume directly leading to a higher cardiac blood volume and therefore a greater use of macro- and micronutrients which is a common cause of high energy demand and expenditure [12]. Apart from which, hypermetabolism in patients with liver cirrhosis may also be related to sympathetic over activity [57]. The hormones of the sympathetic nervous system (SNS) stimulate gluconeogenesis and over time can place the body in a hypermetabolic state, leading to increased muscle breakdown as previously mentioned [23,105]. Results from a cross-sectional study carried out by Müller et al. [105] indicate the presence of hypermetabolism in 34% of patients with liver cirrhosis with differences in REE from predicted values being positively correlated with epinephrine concentration. Therefore, this indicates an activation of the SNS and increased plasma concentrations of catecholamines [68]. In summary, hypermetabolism is a relatively frequent feature in stable liver cirrhosis and is not associated with gender, etiology or severity of liver disease; however, it may result from up-regulation of the SNS [42].

### 2.2. Additional to DRM: Mechanisms Involved in Sarcopenia in Liver Cirrhosis

In the previous sections we addressed the general malnutrition-sarcopenia-syndrome related mechanisms applicable to liver cirrhosis, including increased gluconeogenesis from protein. Having a closer look at the muscle itself, we find additional pathomechanisms promoting the imbalance between muscle formation and muscle breakdown with myostatin as central inhibitor and anabolic hormones as central promoters of muscle growth. These aspects are addressed in the following sections, summarized in Figure 2 below.

Many pathomechanistic insights for liver-disease-associated muscle wasting are derived from rodent models that represent an important resource for preclinical and clinical sarcopenia studies [106]. The known mechanisms contributing to sarcopenia in liver cirrhosis patients are discussed in the following sections.

#### 2.2.1. Inhibition of Muscle Growth and Elevated Myostatin

The best characterized factor contributing to sarcopenia in liver cirrhosis is hyperammonemia [32]. Hyperammonemia mediates the activation of p65-nuclear factor kappa B (NF-kB) which is associated with the transcriptional upregulation and increased expression of myostatin [15,32,107].

Myostatin is a transforming growth factor beta (TGF-β) superfamily member and a critical autocrine/paracrine inhibitor of skeletal muscle growth and mass [108,109]. Experimental studies indicate that mice without myostatin or that were given a molecule inhibiting myostatin expression had increased skeletal muscle mass [11,110]. Furthermore, elevated plasma concentrations of myostatin have been reported in cirrhotic patients compared with controls, demonstrating that myostatin is involved in suppressing skeletal muscle mass in this patient collective [11,30,108]. Pathophysiologically, increased myostatin expression results in a reduction of muscle growth by impairing the mammalian target of rapamycin complex 1 (mTORC1) and its downstream signaling responses in the canonical Akt/mTORC1 pathway [111]. Moreover, increased myostatin production inhibits muscle growth by decreasing satellite cell proliferation and differentiation [32].

Hyperammonemia also contributes to sarcopenia through the liver-muscle axis [32]. The accumulation of ammonia in skeletal muscle prevents the production of α-ketoglutarate, a major substrate for the tricarboxylic acid (TCA) cycle resulting in a lower flux of the TCA cycle, impaired mitochondrial function and decreased synthesis of adenosine triphosphate (ATP). Since protein synthesis, especially translation initiation, is an energy-intense process, low ATP concentrations may also cause reduced protein synthesis [42].

As liver cirrhosis leads to a decline in the capacity of the liver as the key site to detoxify ammonia, the skeletal muscle plays a compensatory role in ammonia metabolism and clearance [112]. The muscle contains the ammonia-removing enzyme glutamine synthetase, compensating ammonia metabolism in chronic liver disease. Therefore, sarcopenia hastens the development of hepatic encephalopathy [113].

##### Hormonal Regulation of Muscle Homeostasis

Furthermore, altered levels of anabolic hormones also play a role in sarcopenia and DRM in liver cirrhosis [29]. Myostatin is typically suppressed by testosterone and insulin-like growth factor 1 (IGF-1) [32]. In liver cirrhosis, decreased levels of testosterone and IGF-1 contribute to increased myostatin expression and impaired protein synthesis [15]. IGF-1 secretion is stimulated by human growth hormone and plays a vital role in many paracrine, autocrine, and endocrine functions. It has multimodal effects on muscle and activates Akt/mTORC1 pathway to stimulate protein synthesis [11,30,114]. Liver cirrhosis is a state of acquired growth hormone resistance. In line with this, low levels of IGF-1 reduce mTOR activation of muscle protein synthesis, further contributing to sarcopenia [32].

Results from a 12-month, double-blinded, placebo-controlled trial in 101 men with established liver cirrhosis and low serum testosterone showed that intramuscular testosterone undecanoate treatment increased muscle mass, bone mass and hemoglobin, and reduced fat mass and HbA1c [115]. Furthermore, effects of daily subcutaneous IGF-1 injections were tested by Conchillo et al. [116] in a randomized controlled trial with 18 liver cirrhosis patients over a period of 4 months. This study revealed, despite increased serum levels of IGF-1 and serum albumin, no changes in body composition, muscle mass or strength measures were detected compared to the placebo group [116]. Nevertheless, a reduction in REE after IGF-1 replacement suggested a role of IGF-1, and the GH axis respectively, in reducing the hypermetabolism associated with cirrhosis. However, the major side effect of GH or IGF-1 administration is fluid retention [114,116].

#### 2.2.2. Increased Muscle Breakdown

Muscle loss is not only a result of the reduction in protein synthesis but also requires increased proteolysis. Muscle depletion is a common complication in liver cirrhosis [112]. Patients with liver cirrhosis have poor glycogen reserves resulting in increased gluconeogenesis and therefore an excessive muscle protein breakdown (proteolysis) resulting in sarcopenia [45].

Likewise, the protein catabolism pathway is a mechanism causing reduced muscle mass, in which mitochondria, lysosomes and the ubiquitin–proteasome system are implicated. The ubiquitin-proteasome system does not only break down abnormal proteins within muscles, but it can also catabolize normal proteins [11,117].

##### Ubiquitin–Proteasome System and Autophagy

Inflammation in chronic disease and circulating proinflammatory cytokines often lead to inappropriate muscle autophagy [32]. Since cytokines emanated by muscles under physiological conditions (myokines) have a range of actions in mediation of inflammation, sarcopenia may therefore potentiate the proinflammatory state of liver cirrhosis that may further reduce muscle mass [15]. In sarcopenia, the major degraded skeletal muscle proteins are myofibrillar proteins, such as myosin heavy chain (MHC), which is a key component in the process of muscle contraction [118]. Sarcopenia in aging as well as chronic liver disease is accompanied by increased muscle autophagy [111]. Muscle autophagy is mediated through increased components of the ubiquitin-proteasome pathway (UPP) which is upregulated by increased levels of proinflammatory cytokines, such as TNF-α and IL-6, as well as reactive oxygen species [32,118]. The ubiquitin-proteasome system (UPS) involves numerous components and its overactivation characterized by a muscle-specific increase of type E3 ubiquitin ligases, atrogin-1 (Muscle Atrophy F-box, MAFbx), and MuRF-1 (Muscle-specific RING-finger protein 1) constitutes a major catabolic mechanism resulting in sarcopenia [118,119].

In liver cirrhosis, hyperammonemia is associated with increased muscle autophagy, too, probably explaining why hepatic encephalopathy is more frequent in sarcopenic than non-sarcopenic cirrhotic patients [20,120].

#### 2.2.3. Physical Activity

Physical activity is an important determinant of muscle anabolism and most patients with cirrhosis, especially those on the transplant waiting lists are mainly sedentary [15]. In patients with liver cirrhosis, regular physical activity is considered an important strategy for preventing, ameliorating or reversing sarcopenia and its complications [20,113]. Consistent with the broader chronic disease literature, the experimental and clinical evidence for a benefit of exercise in liver cirrhosis is promising [19]. Specifically, resistance training in patients with sarcopenia in liver cirrhosis prevents muscle breakdown and maintains physical function through its role in upregulating IGF-1. This upregulation of IGF-1 could result in the downregulation of myostatin; however, more evidence is needed to support this recommendation for this condition [15,32]. Recent randomized controlled trials have shown good tolerance of exercise and improved muscle mass following supervised physical training in cirrhotic patients [113]. However, the optimal type (aerobic vs. resistance) and frequency of exercise programs for liver cirrhosis and hepatic encephalopathy are still unexamined and mechanisms are not fully understood. Additionally, training programs should be combined with timed protein and carbohydrate supply.

## 3. DRM and Sarcopenia Management Strategies Based on Molecular Mechanisms

The molecular mechanisms of DRM and sarcopenia in patients with liver cirrhosis (shown in Figure 1 and Figure 2 above), provide an understanding of possible pathways in addressing these comorbidities. Aforementioned, the molecular mechanisms contributing to DRM and sarcopenia are multifactorial in nature. Therefore, a combination of nutritional, physical and pharmacological interventions might be necessary in addressing these issues [39].

Since energy and protein intake are frequently decreased in liver cirrhosis, increased nutrient intake, high-energy/protein diets, oral nutritional supplements (sip feeds), and when appropriate enteral or parenteral nutrition are often recommended in the treatment of DRM and sarcopenia in liver cirrhosis [32,42]. Several studies have demonstrated the positive effects of BCAA supplementation in malnourished cirrhotic patients with present or previous episodes of hepatic encephalopathy Les et al. [121] in a randomized study that included 116 patients with liver cirrhosis revealed that BCAA supplementation improved minor hepatic encephalopathy and muscle mass. Among the BCAAs, leucine is particularly promising in increasing muscle protein synthesis [19]. Leucine directly activates mTORC1 that stimulates protein synthesis and decreases autophagy, both of which have the potential to improve muscle mass [42]. Preliminary results in patients with alcoholic liver cirrhosis show, that a leucine enriched BCAA mixture is able to reverse the molecular perturbations in the skeletal muscle downstream of myostatin; the impaired mTOR1 signaling and increased autophagy in skeletal muscle of these patients was acutely reversed [122]. In terms of micronutrient replacement, specific evidence about the beneficial effects on muscle mass in cirrhotic patients is unavailable [19]. However, confirmed clinically suspected deficiency should be treated based on accepted guidelines and consensus recommendations [123]. The ability of anabolic hormones to improve muscle mass in liver cirrhosis has been investigated in previous studies [39]. Testosterone and GH have been used to improve nutritional status and potentially muscle mass in liver cirrhosis however, these they proved to be not effective [42]. It is possible that the effects of testosterone may be blunted by increased aromatase activity in this patient group which then enhances the conversion of testosterone to estradiol [19]. Lack of therapeutic benefits with hormone replacement may also be due to impaired signaling responses including mTORC1 response downstream of androgen and GH receptors [42]. Novel approaches, such as myostatin antagonists, direct mTORC1 activators and mitochondrial protective agents, theoretically have potential benefits on the liver-muscle axis, but have not been adequately evaluated in clinical studies [15,19,42]. These novel strategies may hold promise in reversing the molecular abnormalities underlying DRM and sarcopenia in liver cirrhosis [15]. However careful mechanistic studies are necessary with preclinical testing before these interventions can be translated to clinical practice, especially by patients with liver cirrhosis [19,42].

## 4. Conclusions

In the setting of liver disease, DRM and sarcopenia are considered as common and significant complications of liver cirrhosis associated with adverse clinical outcomes. However, with an expanding scientific evidence addressing DRM and sarcopenia in liver cirrhosis, there are still inadequate patient data regarding specific molecular mechanisms contributing to these comorbidities. This could possibly be because of overlaps between different molecular pathways and entities that are interconnected, multifactorial and complex in nature. On one hand, there are no precise mechanistic data in the case of SIBO effects in patients with liver cirrhosis. While data pertaining to leptin, PYY, CCK and GLP-1 concentrations are inconsistent and limited. Data considering hypermetabolism are equally limited with barely any updated research in this area. The proposed molecular mechanisms contributing to DRM and sarcopenia provide a better understanding on the mechanisms underlying nutritional deficiencies, hormonal abnormalities, increased energy and protein requirements, altered metabolic pathways, inflammation, hyperammonemia, muscle breakdown and inactivity in patients with liver cirrhosis. More research is, therefore, needed in order to implement and recommend optimal therapies aimed at addressing DRM and sarcopenia in patients with liver cirrhosis.

## Figures and Tables

**Figure 1 ijms-21-05357-f001:**
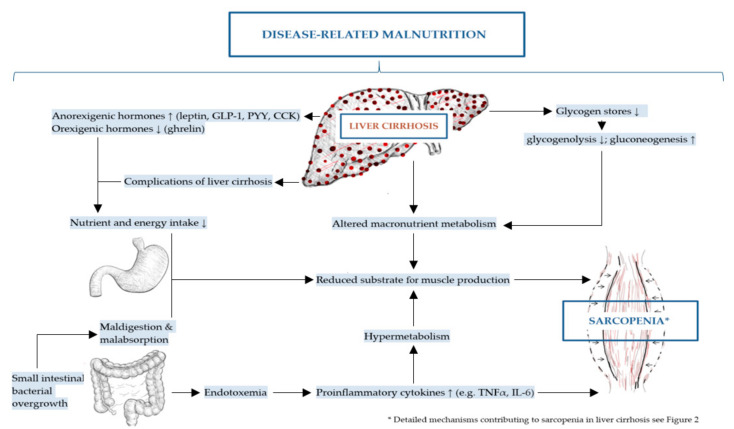
Mechanisms contributing to disease-related malnutrition (DRM) and sarcopenia in liver cirrhosis.

**Figure 2 ijms-21-05357-f002:**
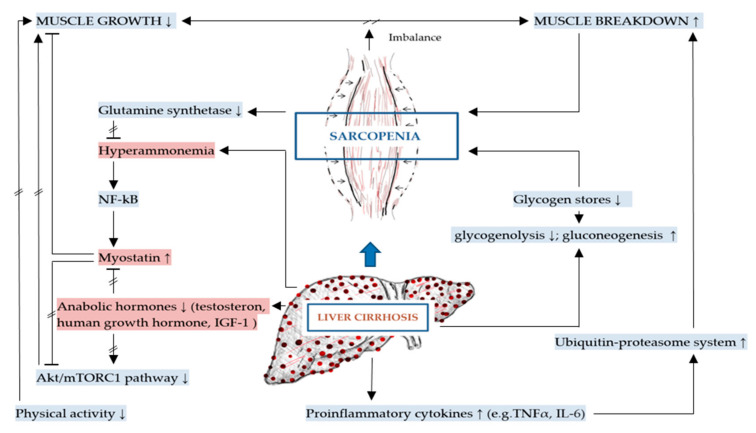
Mechanisms specific to sarcopenia in liver cirrhosis (key mechanisms contributing to sarcopenia in liver cirrhosis are highlighted in red).

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
