# Peer review of "Molecular Mechanism Contributing to Malnutrition and Sarcopenia in Patients with Liver Cirrhosis"

_ijms, 2020, doi:10.3390/ijms21155357_

Round 1

Reviewer 1 Report

A very good manuscript. The authors present state of art. The practical part is missing. I am asking for the addition of some therapeutic methods that inhibit the pathophysiology of sarcopenia in the course of liver cirrhosis. The manuscript will gain importance. How do the therapeutic methods influence on the pathophysiology of sarcopenia in the course of liver cirrhosis? 

Author Response

We thank the reviewer for the constructive comment and added the following paragraph to the manuscript:

  1. DRM and Sarcopenia management strategies based on molecular mechanisms

                The molecular mechanisms of DRM and sarcopenia in patients with liver cirrhosis (shown in Figure 1 and 2 above), provide an understanding of possible pathways in addressing these comorbidities. Aforementioned, the molecular mechanisms contributing to DRM and sarcopenia are multifactorial in nature. Therefore, a combination of nutritional, physical, and pharmacological interventions might be necessary in addressing these issues [39].         

                Since energy and protein intake are frequently decreased in liver cirrhosis, increased nutrient intake, high-energy/-protein diets, oral nutritional supplements (sip feeds), and, when appropriate enteral or parenteral nutrition  are often recommended in the treatment of DRM and sarcopenia in liver cirrhosis [32, 42]. Several studies have demonstrated the positive effects of BCAA supplementation in malnourished cirrhotic patients with present or previous episodes of hepatic encephalopathy Les et al. [121] in a randomized study that included 116 patients with liver cirrhosis revealed that BCAA supplementation improved minor hepatic encephalopathy and muscle mass. Among the BCAAs, leucine is particularly promising in increasing muscle protein synthesis [19]. Leucine directly activates mTORC1 that stimulates protein synthesis and decreases autophagy, both of which have the potential to improve muscle mass [42]. Preliminary results in  patients with alcoholic liver cirrhosis show, that a leucine enriched BCAA mixture is able to reverse the molecular perturbations in the skeletal muscle downstream of myostatin; the impaired mTOR1 signalling and increased autophagy in skeletal muscle of these patients was acutely reversed [122]. In terms of micronutrient replacement, specific evidence about the beneficial effects on muscle mass in cirrhotic patients is unavailable [19]. However, confirmed clinically suspected deficiency should be treated based on accepted guidelines and consensus recommendations [123]. The ability of anabolic hormones to improve muscle mass in liver cirrhosis has been investigated in previous studies [39]. Testosterone and GH have been used to improve nutritional status and potentially muscle mass in liver cirrhosis however, these they proved to be not effective [42]. It is possible that the effects of testosterone may be blunted by increased aromatase activity in this patient group which then enhances the conversion of testosterone to estradiol [19]. Lack of therapeutic benefits with hormone replacement may also be due to impaired signaling responses including mTORC1 response downstream of androgen and GH receptors [42]. Novel approaches such as myostatin antagonists, direct mTORC1 activators, and mitochondrial protective agents theoretically have potential benefits on the liver‐muscle axis, but have not been adequately evaluated in clinical studies [15, 19, 42]. These novel strategies may hold promise in reversing the molecular abnormalities underlying DRM and sarcopenia in liver cirrhosis [15]. However careful mechanistic studies are necessary with preclinical testing before these interventions can be translated to clinical practice, especially by patients with liver cirrhosis [19, 42].”

L472: New reference 121

  1. Les, I.;Doval, E.;García-Martínez, R.;Planas, M.;Cárdenas, G.;Gómez, P.;Flavià, M.;Jacas, C.;Mínguez, B.;Vergara, M.; et al. Effects of branched-chain amino acids supplementation in patients with cirrhosis and a previous episode of hepatic encephalopathy: a randomized study. Am J Gastroenterol. 2011, 106, 1081-1088. DOI: 10.1038/ajg.2011.9.

L479: New references 122

  1. Tsien, C.;Davuluri, G.;Singh, D.;Allawy, A.;Ten Have, G.A.;Thapaliya, S.;Schulze, J.M.;Barnes, D.;McCullough, A.J.;Engelen, M.P.; et al. Metabolic and molecular responses to leucine-enriched branched chain amino acid supplementation in the skeletal muscle of alcoholic cirrhosis. Hepatology. 2015, 61, 2018-2029. DOI: 10.1002/hep.27717.

L482: New references 123

  1. European Association for the Study of the Liver. EASL Clinical Practice Guidelines on nutrition in chronic liver disease. J Hepatol. 2019, 70, 172-193. DOI: 10.1016/j.jhep.2018.06.024.

Reviewer 2 Report

Meyer F et al., Molecular Mechanism Contributing to Malnutrition and Sarcopenia in Patients with Liver Cirrhosis

In this review article, Meyer et al precisely reviewed the molecular mechanisms implicated in malnutrition and sarcopenia in cirrhotic patients. The references have been comprehensively searched for and the quality of the figures is fine. However, the following is suggested:

It would be nicer if the authors can present any currently applied or potentially feasible therapy based upon the molecular mechanisms implicated in malnutrition and sarcopenia of cirrhotic patients, just before the section of Conclusion.

Author Response

(The authors gave the same response as above.)
